# Determinants of post-acute COVID-19 syndrome among hospitalized severe COVID-19 patients: A 2-year follow-up study

Tamrat Petros Elias[ID][1]*, Tsegaye Wesenseged Gebreamlak[2], Tigist Tesfaye Gebremeskel[3], Binyam Lukas Adde[4], Abraham Sisay Abie[1], Bitaniya Petros Elias[5], Abel Mureja Argaw[ID][1], Addis Aschenek Tenaw[ID][6], Biruk Mulugeta Belay[1]

1 Department of Internal Medicine, St. Paul's Hospital Millennium Medical College, Addis Ababa, Ethiopia, 2 Department of Internal Medicine, Adera Medical and Surgical Center, Addis Ababa, Ethiopia, 3 Department of Pediatrics, St. Peter's Specialized Hospital, Addis Ababa, Ethiopia, 4 Department of Internal Medicine, Menelik II Specialized Hospital, Addis Ababa, Ethiopia, 5 Department of Research and Development, Deborah Foundation, Addis Ababa, Ethiopia, 6 Department of Internal Medicine, Yekatit 12 Medical College, Addis Ababa, Ethiopia

* tamrat.petros@sphmmc.edu.et

**Data Availability Statement:** All raw data files are available from the Dryad repository database (https://datadryad.org/stash/share/X4rfM77vUc2yRwvli0mI9FiyXbIla0CzHgJJdwJTv5g).

## Abstract

### Background

Post-acute COVID-19 syndrome is a condition where individuals experience persistent symptoms after the acute phase of the COVID-19 infection has resolved, which lowers their quality of life and ability to return to work. This study assessed the prevalence and associated risk factors of post-acute COVID-19 syndrome (PACS) among severe COVID-19 patients who were discharged from Millennium COVID-19 Care Center, Addis Ababa, Ethiopia.

### Methods

A cross-sectional study using data collected from patient charts and a follow-up telephone interview after two years of discharge. Systematic random sampling was used to select a total of 400 patients. A structured questionnaire developed from the case report form for PACS of the World Health Organization (WHO) was used. Frequency and cross-tabulation were used for descriptive statistics. Predictor variables with a p-value <0.25 in bivariate analyses were included in the logistic regression.

### Result

Out of the 400 patients, 20 patients were dead, 14 patients refused to give consent, and 26 patients couldn't be reached because their phones weren't working. Finally, 340 were included in the study. The majority (68.5%) were males and the mean age was 53.9 (±13.3 SD) years. More than a third (38.1%) of the patients reported the persistence of at least one symptom after hospital discharge. The most common symptoms were fatigue (27.5%) and Cough (15.3%). Older age (AOR 1.04, 95% CI 1.02–1.07), female sex (AOR 1.82, 95% CI 1.00–3.29), presence of comorbidity (AOR 2.38, 95% CI 1.35–4.19), alcohol use (AOR

**Funding:** The author(s) received no specific funding for this work.

**Competing interests:** The authors have declared that no competing interests exist.

3.05, 95% CI 1.49–6.26), fatigue at presentation (AOR 2.18, 95% CI 1.21–3.95), and longer hospital stay (AOR 1.06, 95% CI 1.02–1.10) were found to increase the odds of developing post-acute COVID-19 syndrome. Higher hemoglobin level was found to decrease the risk of subsequent post-acute COVID-19 syndrome (AOR 0.84, 95% CI 0.71–0.99).

## Conclusion

The prevalence of post-acute COVID-19 syndrome is high, with a wide range of persistent symptoms experienced by patients. COVID-19 survivors with the identified risk factors are more susceptible to post-acute COVID-19 and require targeted monitoring and care in a multidisciplinary approach.

## Introduction

SARS-CoV-2, which causes coronavirus disease-19 (COVID-19), emerged as a public health threat in December 2019 [1]. According to the online World Health Organization (WHO) COVID-19 dashboard, as of May 1, 2023, the COVID-19 pandemic affected more than 765 million people and caused more than 6.9 million deaths globally [2]. The detection and treatment of acute illness does not appear to be the end of the COVID-19 fight. It has lately come to light that some patients' incapacitating symptoms might last for weeks or even months [3]. This manifestation was termed 'post-acute COVID-19 syndrome', 'post COVID conditions', 'chronic COVID-19', or 'long COVID' [4]. The number of post-acute COVID-19 patients is rapidly increasing because millions of people have already contracted the disease and many more will do so in the future [5]. The capacity of people to return to work can be seriously impacted by persistent COVID-19 symptoms, with substantial psychological, social, and economic repercussions for those affected, their families, and society as a whole [6]. The annual economic impact of PACS (exclusive of costs of disability services, social services, and lost income on the part of caretakers) in the United States ranges from $140 billion to $600 billion [7].

The pathophysiology of PACS is multi-factorial and more than one mechanism may be implicated in several clinical manifestations. Immune dysregulation, persistent inflammatory reactions, autoimmune mimicry, pathogen reactivation, and host-microbiome changes may all play a role in the development of the syndrome [8]. Its pathophysiology is significantly influenced by prolonged inflammation, which can also be the cause of other symptoms such as cognitive impairment and neurological problems. Similar to multisystem inflammatory syndrome in children (MIS-C), a multisystem inflammatory syndrome in adults (MIS-A) of all ages has also been recently described [9].

The prevalence and clinical presentation of PACS is highly heterogeneous. The most frequently reported symptoms are fatigue, cardio-respiratory problems, and neurological symptoms [10]. There is a wide difference in the prevalence of post-acute COVID-19, from 46% in Bangladesh to 87.4% in Italy [11,12]. Some researchers concluded that female gender and older age are important risk factors for eventual PACS [13,14], but others, found no link between these sociodemographic characteristics and the development of PACS [15]. There is a significant difference in the works of literature on whether or not risk factors for developing PACS include the existence of comorbidities [13,16], the type of symptoms that present during an acute illness [15,17], the length of hospitalization [11,15], and the amount of oxygen needed upon admission [18]. Cigarette smoking was not associated with PACS in some studies [19], while others found a strong association [20].

Ethiopia notified the first confirmed case of COVID-19 on March 13, 2020 [21]. The country recorded the largest number of COVID-19 confirmed cases in East Africa [2], implying a large number of patients with PACS. There are no post-COVID clinics in Ethiopia, nor is there a documented guideline for the management of post-COVID sequelae. There were no research articles in peer-reviewed journals measuring the burden of PACS in Ethiopia at the time this study was conceived. To alleviate these issues, we need to understand the prevalence and risk factors of PACS to establish effective management measures such as rehabilitation and support services. This may include physical therapy, occupational therapy, cognitive interventions, and mental health support to address the diverse range of symptoms and disabilities experienced by individuals with PACS.

As a result, this study aimed to assess the prevalence and associated risk factors of PACS among severe COVID-19 patients who were discharged alive from Millennium COVID-19 Care and Treatment Center, Addis Ababa, Ethiopia between June 12, 2020, and November 1, 2021.

## Methodology

### Study design and setting

A cross-sectional study design was used to assess the prevalence and associated risk factors of post-acute COVID-19 syndrome among severe COVID-19 patients who were discharged alive from Millennium COVID-19 Care Center, Addis Ababa, Ethiopia. Millennium COVID-19 Care Center (MCCC), was a makeshift hospital in Addis Ababa, the capital city of Ethiopia. The center was the biggest COVID-19 treatment facility in the country. It began giving service on June 2, 2020, and according to the center's health management information system report (HMIS), as of November 1, 2021, a total of 6,760 patients were admitted and 5,580 patients were discharged alive.

### Study participants

From adult patients (>18 years of age) who were admitted to MCCC with the diagnosis of Severe COVID-19 infection, confirmed by polymerase chain reaction (PCR) or rapid diagnostic test (RDT), those who were discharged alive between June 12, 2020, and November 1, 2021, were the study population.

### Data collection tools and procedures

The socio-demographic profiles, past medical history including comorbidity, duration of symptoms before hospital admission, length of hospital stay, the maximum amount of oxygen required during hospital stay, acute manifestations of COVID-19, and baseline laboratory investigations were extracted from patient charts from January 2, 2023, to January 31, 2023. After obtaining verbal consent, a detailed telephonic interview was conducted with the study participants between February 1, 2023, and April 30, 2023, to record self-reported PACS symptoms and their characteristics, and self-assessment of current health status compared to the pre-COVID state. Data on COVID-19 vaccination history and current substance use were also collected during the telephone interview. The questionnaire was adapted from the W.H.O Global COVID-19 Clinical Platform Case Report Form (CRF) for Post COVID conditions (Post COVID-19 CRF) [22]. The English version of the questionnaire was first translated into Amharic, and a pilot study was done on 20 patients discharged from Eka Kotebe COVID-19 treatment center. A minor revision was made to the structure and language of the translated version based on the feedback from the participants in the pilot study. Data collectors were

given training before the data collection. The collected data were entered into Epi-info software version 7 and then exported to Statistical Package for Social Sciences (SPSS) version 25 for cleaning and analysis. Individuals who could not be contacted after two attempts were excluded.

## Sample size and statistical analysis

The sample size was determined for the prevalence by using the single population proportion formula and for the sociodemographic, clinical, and behavioral risk factors by using the double population proportion formula. The sample size calculated for the prevalence by using the single population proportion formula by considering p = 50%, as the prevalence of post-acute COVID syndrome is not known in Ethiopia, 95% confidence level ($Z\alpha/2 = 1.96$), and a 5% margin of error yields the largest sample size. Since the source population is less than ten thousand (6760), a population correction formula was used and a 10% non-response rate was added to yield a final sample size of 400.

The medical record chart of patients who were admitted to the MCCC, along with the HMIS register book, is stored in an isolated corner within the medical record room of SPHMMC (St. Paul's Hospital Millennium Medical College). Of the 5580 patients discharged alive from the center during the study period, 3576 of them were admitted with the diagnosis of Severe COVID-19. To obtain a representative sample, the sampling procedure followed a systematic random sampling. In systematic random sampling, the value of k (the sampling interval) is determined by dividing the population size (N) by the desired sample size (n) and rounding to the nearest whole number. The formula is as follows:

$$k = N/n$$

In this study, N (Study population) is 3576, and n (desired sample size) is 400. Using the above formula, k (sampling interval) is calculated to be 9. Every ninth patient discharged alive from MCCC with an admission diagnosis of severe COVID-19 and meeting the inclusion criteria is included in the study.

Frequency and cross-tabulation are used to summarize descriptive statistics of the data. The Mann–Whitney U test was used to compare skewed continuous variables. Associations between predictor variables and outcomes of interest are estimated using both bivariate analysis and binary logistic regression. Predictor variables with a p-value <0.25 in bivariate analyses are reported and included in the logistic regression. For the Binary Logistic regression, a 95% confidence interval for adjusted odds ratio (AOR) was calculated and variables with p-value ≤ 0.05 were considered as statistically associated with PACS.

## Operational definitions

**Severe COVID-19.**   Patients with clinical signs of severe pneumonia, ARDS, or sepsis ANDoxygen saturation less than 90% on room air; OR respiratory rate greater than 30.

**Post-acute COVID-19 syndrome.**   Persistence of any sign or symptom that was developed during the acute COVID-19 illness for more than twelve weeks after hospital discharge.

## Ethical consideration

The study was conducted after obtaining ethical clearance from St. Paul's Hospital Millennium Medical College Institutional Review Board. Verbal informed consent was taken from study participants during the telephone interview and documented on the questionnaire after explaining the purpose and objectives of the study. Confidentiality of individual patient

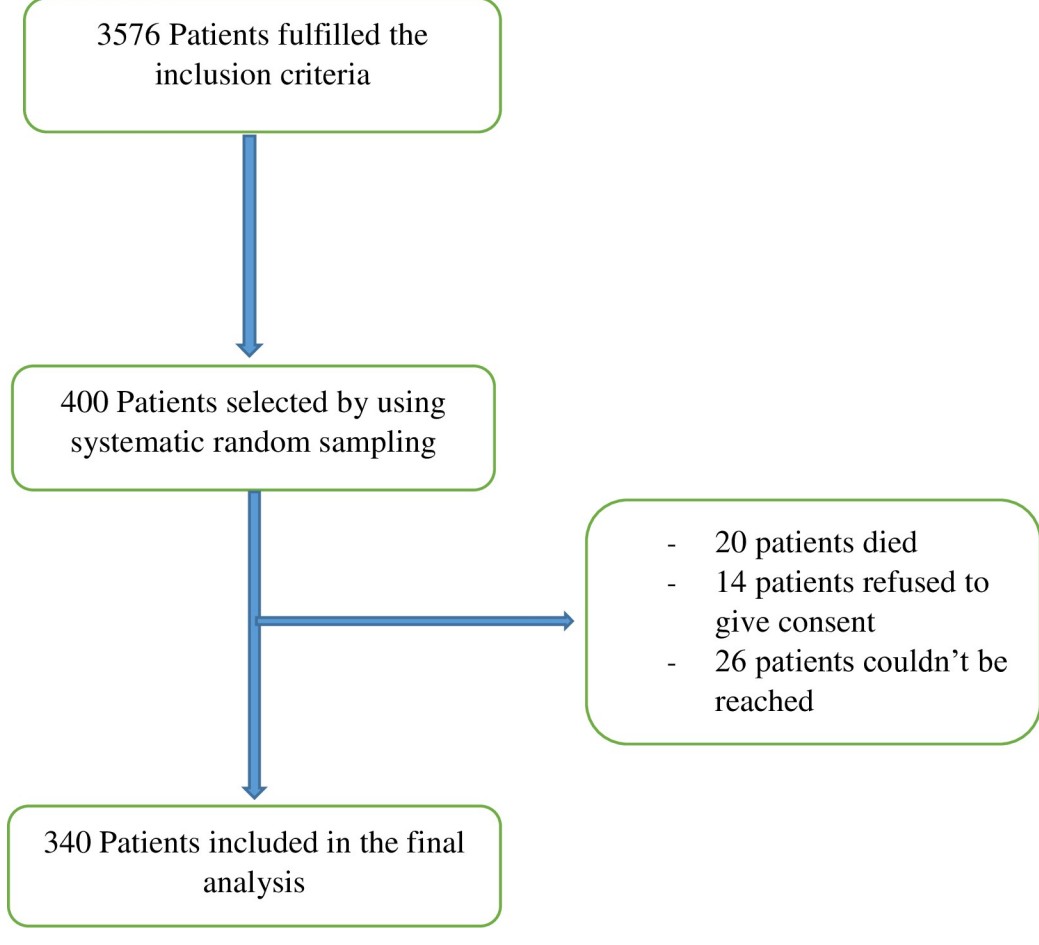

**Fig 1. Patient selection for this study.**

information is maintained by using code numbers instead of other identifiers and the information gained from the chart and phone call is used only for research purposes.

## Results

Out of the 400 patients selected for the study, 20 patients (5%) died after hospital discharge, 14 patients (3.5%) refused to give consent, and 26 patients (6.5%) couldn't be reached because their phone wasn't working. The study included a total of 340 patients who were admitted to Millennium COVID-19 Care and Treatment Center with the diagnosis of severe COVID-19 pneumonia and discharged alive between June 12, 2020, and November 1, 2021 (Fig 1). The mean duration from hospital discharge to the interview was 25.6 (± 4.8) months.

### Baseline sociodemographic and clinical characteristics

The majority (68.5%) of the study participants were male and the remaining 31.5% were females. The mean age at the time of admission to the center was 53.9 (±13.3 SD) years. The minimum age was 22 years and the maximum age was 85 years.

More than half of the patients (60%) have one or more comorbidities. As shown in Table 1, the most common comorbidity among the patients was diabetes (35.6%), followed by

**Table 1. Comorbidity pattern of patients.**

| Comorbidity | Frequency (Percent) |
| --- | --- |
| Diabetes | 121 (35.6%) |
| Hypertension | 116 (34.1%) |
| Chronic heart disease | 22 (6.5%) |
| Asthma or COPD | 19 (5.6%) |
| Dyslipidemia | 13 (3.8%) |
| HIV | 11 (3.2%) |
| Cancer | 5 (1.5%) |
| Chronic liver disease | 5 (1.5%) |
| Stroke | 4 (1.2%) |
| Chronic kidney disease | 1 (0.3%) |

hypertension (34.1%), chronic heart disease (6.5%), asthma or COPD (5.6%), dyslipidemia (3.8%), HIV (3.2%), Cancer (1.5%), CLD (1.5%), stroke (1.2%), and CKD (0.3%).

The median (IQR) duration of symptom onset before admission was 7 (5–10) days. The minimum was 1 day and the maximum was 30 days. The median (IQR) duration of hospital admission was 11 (7–15.75) days. The minimum was 2 days and the maximum was 39 days. The median (IQR) maximum amount of oxygen required during a hospital stay was 5 (3–7) liters. The minimum oxygen requirement was 1 liter and the maximum was 15 liters. A Mann-Whitney U test was conducted to compare the median scores of patients with and without PACS on duration of symptom onset before admission, length of hospital stay, and maximum amount of oxygen required during hospital stay. The test was statistically significant ($p < 0.01$) only for the length of hospital stay.

As shown in Table 2, the most common symptoms at presentation were cough (93.5%), followed by shortness of breath (82.1%), and fatigue (69.7%).

In the complete blood count (CBC) parameter of the patients, there were two outlier results recorded on patients with chronic lymphocytic leukemia (CLL), who had the white blood cell (WBC) count of 350,000 cells/ μL and 49,000 cells/ μL. The mean WBC count after excluding these two results was 9,148 (±4,044 SD) cells/ μL. The patients had a low mean lymphocyte percentage (10.2%). The mean hemoglobin was 14.6 (±1.7 SD) gm/dl and the mean platelet count was 285,840 (±116,588 SD) cells/ μL.

**Table 2. Presenting symptoms.**

| Symptom | Frequency (Percent) |
| --- | --- |
| Cough | 318 (93.5%) |
| Shortness of breath | 279 (82.1%) |
| Fatigue | 237 (69.7%) |
| Fever | 181 (53.2%) |
| Joint pain | 166 (48.8%) |
| Headache | 122 (35.9%) |
| Chest pain | 103 (30.3%) |
| Loss of appetite | 88 (25.9%) |
| Diarrhea | 79 (23.2%) |
| Loss of taste | 27 (7.9%) |
| Loss of smell | 12 (3.5%) |

**Table 3. Symptoms that have persisted after hospital discharge.**

| Symptom | Frequency (Percent) |
| --- | --- |
| Fatigue | 88 (27.5%) |
| Cough | 49 (15.3%) |
| Joint pain | 45 (14.1%) |
| Headache | 38 (11.9%) |
| Shortness of breath | 36 (11.3%) |
| Sleep disturbance | 35 (10.9%) |
| Chest pain | 18 (5.6%) |
| Loss of appetite | 18 (5.6%) |
| Loss of smell | 18 (5.6%) |
| Diarrhea | 8 (2.5%) |
| Loss of taste | 8 (2.5%) |

## Current status of study participants

Most (67.8%) of study participants visited a health facility at least once after their discharge from Millennium COVID-19 care and treatment center. The most common reason (59%) for the health facility visit was for follow-up of chronic disease and 15.7% of the reasons for hospital visit after discharge were not feeling well.

More than a third (38.1%) of the patients reported the persistence of at least one symptom after hospital discharge. As shown in Table 3, the most common symptoms that started during the acute COVID-19 infection and continued till the time of the interview in descending order were; fatigue (27.5%), Cough (15.3%), joint pain (14.1%), headache (11.9%), and shortness of breath (11.3%). Symptoms that were less commonly found were diarrhea (2.5%) and loss of appetite (2.5%).

Close to one-third (29.4%) of the patients feel that their health condition has deteriorated after the COVID-19 infection. Forty-four (13.8%) of the patients claimed that, currently they are not able to do the daily activities they used to do before the infection which forced some of the patients to change and even quit their jobs. Currently, three (0.9%) and fifty-five (17.2%) of the patients smoke cigarettes and drink alcohol respectively. Of those who drink alcohol, most (40%) drink twice per month, followed by once per week (18.2%). Seven patients (12.7%) drink alcohol daily. Only 28.4% of the patients received at least one dose of vaccination.

## Factors associated with PACS

Patient age, sex, presence of comorbidity, alcohol use, baseline hemoglobin level, initial presentation with fatigue or loss of appetite, and length of hospital stay were found to be associated with the development of PACS in the patients at a significant level of $P < 0.05$. However, cigarette smoking, current vaccination status, mean day of presentation after symptom onset, and maximum amount of oxygen used during the hospital stay were not found to influence the development of post-acute COVID-19 syndrome (Table 4).

By using variables that have a p-value of $<0.25$ in the bivariate analysis, binary logisticregression was done after the model fitness test. Factors that were independently associated with the development of PACS were older age (AOR 1.04, 95% CI 1.02–1.07), female sex (AOR 1.82, 95% CI 1.00–3.29), presence of comorbidity (AOR 2.38, 95% CI 1.35–4.19), alcohol use (AOR 3.05, 95% CI 1.49–6.26), fatigue at presentation (AOR 2.18, 95% CI 1.21–3.95), and longer hospital stay (AOR 1.06, 95% CI 1.02–1.10). Higher hemoglobin level was found to decrease the risk of subsequent PACS (AOR 0.84, 95% CI 0.71–0.99) (Table 5).

**Table 4. Cross-tabulation of sociodemographic, clinical, and behavioral patterns with PACS.**

| Characteristics | | Has symptom | Has no symptom | P Value |
|---|---|---|---|---|
| Mean Age (years) | | 58.7 | 49.8 | < 0.01 |
| Sex | | | | 0.01 |
| | Male | 74 (33.6%) | 146 (66.4%) | |
| | Female | 48 (48%) | 52 (52%) | |
| Comorbidity | | | | < 0.01 |
| | Yes | 91 (48.4%) | 97 (51.6%) | |
| | No | 31 (23.5%) | 101 (76.5%) | |
| Specific comorbidity | | | | |
| | Hypertension | 56 (53.3%) | 49 (46.7%) | <0.01 |
| | Diabetes | 48 (41.1%) | 68 (58.6%) | 0.36 |
| | Dyslipidemia | 8 (66.7%) | 4 (33.3%) | 0.03 |
| | Chronic kidney disease | 1 (100%) | 0 (0%) | 0.20 |
| | Chronic liver disease | 3 (75%) | 1 (25%) | 0.12 |
| | Chronic heart disease | 14 (87.5%) | 2 (12.5%) | <0.01 |
| | Cancer | 4 (100%) | 0 (0%) | 0.01 |
| | HIV | 6 (60%) | 4 (40%) | 0.14 |
| | Stroke | 2 (66.7%) | 1 (33.3%) | 0.30 |
| | Asthma or COPD | 8 (50%) | 8 (50%) | 0.31 |
| Cigarette smoking | | 0 (0%) | 3 (100%) | 0.17 |
| Alcohol use | | 28 (50.9%) | 27 (49.1%) | 0.03 |
| Mean length of hospital stay (days) | | 14.0 | 11.2 | <0.01 |
| Symptom duration before admission | | 8.5 | 8.3 | 0.80 |
| Mean maximum amount of oxygen | | 6.2 | 5.6 | 0.15 |
| Mean baseline hemoglobin | | 14.3 | 14.8 | <0.01 |
| Current vaccination status | | | | 0.34 |
| | Vaccinated at least once | 31 (34.1%) | 60 (65.9%) | |
| | Not vaccinated | 91 (39.7%) | 138 (60.3%) | |
| Fatigue at presentation | | 93 (41.9%) | 129 (58.1%) | 0.03 |
| Loss of appetite at presentation | | 24 (28.6%) | 60 (71.4%) | 0.03 |

## Discussion

As to the Authors' knowledge, this is the first study that assessed the health consequences of COVID-19 at a two-year follow-up in patients who had severe COVID-19 pneumonia. 340 patients who were admitted to Millennium COVID-19 care and treatment center with the

**Table 5. Crude and adjusted odds ratio of factors that have a significant association with PACS.**

| Characteristics | COR (95% CI) | P-Value | AOR(95%CI) | P-Value |
|---|---|---|---|---|
| Age in years | 1.05 (1.03–1.08) | <0.01 | 1.04 (1.02–1.07) | <0.01 |
| Female sex | 1.82 (1.12–2.95) | 0.01 | 1.82 (1.00–3.29) | 0.04 |
| Presence of comorbidity | 3.05 (1.86–5.01) | <0.01 | 2.38 (1.35–4.19) | <0.01 |
| Drinking alcohol | 1.88 (1.05–3.38) | 0.03 | 3.05 (1.49–6.26) | <0.01 |
| Length of hospital stay | 1.06 (1.03–1.10) | <0.01 | 1.06 (1.02–1.10) | 0.01 |
| Hemoglobin level | 0.82 (0.71–0.95) | <0.01 | 0.84 (0.71–0.99) | 0.04 |
| Fatigue at presentation | 1.71 (1.03–2.85) | 0.03 | 2.18 (1.21–3.95) | <0.01 |
| Loss of appetite | 0.56 (0.32–0.96) | 0.03 | 0.59 (0.31–1.09) | 0.09 |

diagnosis of severe COVID-19 pneumonia were included in the study. The majority (68.5%) were males and the mean age was 53.9 (±13.3 SD) years. More than a third (38.1%) of the patients reported the persistence of at least one symptom after hospital discharge. The most common symptoms were fatigue (27.5%) and Cough (15.3%). Older age, female sex, presence of comorbidity, alcohol use, low baseline hemoglobin level, fatigue at presentation, and prolonged hospital stay were found to increase the odds of developing PACS. Thus, the study found that a significant proportion of patients don't completely recover and continue to have some of the symptoms they developed during the acute infection.

The patients in this study were younger and males were more represented when compared to other similar studies. The mean age (53.9 years) at presentation was younger by six years in this study when compared to the cohort study conducted in Italy [20]. This is probably due to the demographic background of Ethiopia, where the proportion of the elderly population is lower than that in the Western world. The male-to-female ratio was 1.4:1 in the Bangladesh study, but the ratio is significantly higher (2.1:1) in this study [11].

The death rate after hospital discharge in this study was lower (5.8%) compared to a study done in Spain on patients who were admitted to Hospital for COVID-19. The study done in Spain found that 7.5% of the patients died within a mean follow-up period of one year [23].

The study participants in this study have a significantly higher level of comorbidity (60%) when compared to the cross-sectional study in Egypt on 430 patients found that 26.5% of patients reported that they have a chronic illness, and the Norwegian prospective cohort study, in which, 44% had comorbidities [16,24]. This significant difference in comorbidity is observed mainly because the other studies were done on all COVID-19 patients, and this study was done specifically on patients with severe COVID-19 pneumonia. Diabetes (35.6%), hypertension (34.1%), and chronic heart disease (6.5%) were the most common comorbidities in this study which is similar to the cohort studies done in Romania and England [25,26].

The prevalence of PACS in this study was 38.1%, which is lower than the finding in most of the studies, which were in the range of 46% in Bangladesh to 87.4% in Italy [11,12]. This may be because, in those studies, the maximum follow-up period was one year, but this study was conducted after a mean period of 25.6 months after hospital discharge and symptoms might have improved over time. The causes underlying these persistent symptoms following COVID-19 are not entirely understood. In addition to the direct effects of SARS-CoV-2, the immunological response to the virus is thought to have a role in the development of these long-term symptoms, presumably by supporting a continuing hyper-inflammatory process [27]. Molecular hydrogen inhalation had beneficial health effects in terms of improved physical (6-min walking test) and respiratory function in patients with PACS. Patients also noticed an improvement in fatigue after undergoing hyperbaric oxygen therapy and enhanced external counterpulsation. Muscle strength and physical function were improved after undergoing an 8-week biweekly physical therapy course including aerobic training, strengthening exercises, and diaphragmatic breathing techniques [28].

Fatigue is the most common (27.5%) symptom of PACS in this study, which is in concordance with the findings of most other similar studies [12,15,16]. Although the exact cause and pathogenesis of fatigue following COVID-19 is unknown, previous data from severe acute respiratory syndrome (SARS) suggests that lung diffusion capacity impairment, some extrapulmonary causes, such as viral-induced myositis at initial presentation, cytokine disturbance, muscle wasting, and deconditioning, or corticosteroids myopathy, or a combination of these factors, may have contributed to the condition [29].

The second most common symptom in this study was cough (15.3%). This finding mirrors the findings of previous similar studies [11,19,30]. The mechanisms of cough after COVID-19

are multifactorial, including parenchymal sequelae and activation of the vagal sensory nerves, which leads to a cough hypersensitivity state [29].

In this study, 48% of female patients reported the presence of symptoms at the time of the interview. The female sex (AOR = 1.82, 95% CI 1.00–3.29, P = 0.04) was found to increase the risk of developing PACS. These findings are similar to other studies [13,14]. Various underlying processes explaining why females experience post-COVID symptoms to a larger extent than males are now being studied in the literature. Male and female biological differences in the expression of angiotensin-converting enzyme-2 (ACE2) and transmembrane protease serine 2 (TMPRSS2) receptors, as well as immunological differences, such as lower production of pro-inflammatory interleukin-6 (IL-6) after viral infection in females, could explain the higher development of post-COVID symptoms [31].

Older age (AOR = 1.04, 95% CI 1.02–1.07, P: <0.01) was found to be a statistically significant predictor for the development of PACS. This is a similar finding to a study done in France where older age increased the risk of subsequent PACS (AOR = 1.49, 95% CI 1.05–2.17) [17].

Unlike the Bangladesh cohort study which showed patients with fever, cough, respiratory distress, and lethargy as the presenting features were more susceptible to develop PACS compared to other presenting features and the Indian study which showed diarrhea at presentation to be associated with PACS, the only presenting feature that was found to be significant in this study was fatigue (AOR = 2.18, 95% CI 1.21–3.95, P: <0.01) [11,32].

In this study, prolonged hospital stay was found to significantly increase the risk of PACS (AOR = 1.06, 95% CI 1.02–1.10, P = 0.01). A similar finding was observed in a study conducted in Spain which revealed that the number of days at the hospital was significantly associated with an increased risk of PACS [33].

Although the number of days between symptom onset and admission in the Indian study and the amount of oxygen requirement in the Egyptian study was found to determine PACS, this study found no association between those factors and PACS [18,32].

Another finding in our study was that COVID-19 vaccination was not found to be protective from PACS. This was also shown in previous studies [34].

The studies on the effect of smoking as a risk factor for developing post-acute COVID-19 syndrome showed conflicting results. A study conducted in Egypt showed that there is no significant association between cigarette smoking and post-acute COVID-19 syndrome [18], while a study conducted in Italy found a strong association between current active smoking and post-acute COVID-19 syndrome [20]. This study found no significant association between current cigarette smoking and PACS.

The effect of alcohol intake on the development of PACS is found to be significant (AOR = 3.05 [1.49–6.26], P: <0.01) in this study, which is a similar finding to the Mediterranean cohort study and the Bangladesh study [11,15].

The study has certain limitations. This is a single-center cross-sectional study which makes generalization of the findings from this research difficult. The findings of this study might have also suffered from the fact that it is done via a telephone interview which relies on self-reporting and can be subject to recall bias. Furthermore, incomplete documentation on patient charts is another limitation. This study did not consider the broad range of patient characteristics because of the paucity of data. Mainly, nutritional status which was included in most other studies isn't included in the analysis of this research because of incomplete documentation in most of the charts. Future research should consider longitudinal multi-center studies to enhance generalizability, employ diverse data collection methods, and use a larger sample size to ensure the robustness of the findings.

## Conclusion

The prevalence of PACS syndrome among severe COVID-19 patients who were discharged alive from Millennium COVID-19 Care and Treatment Center between June 12, 2020, and November 15, 2021, after a mean period since discharge of 25.6 months, was found to be 38.1%. Fatigue (27.5%) and cough (15.3%) were the most prevalent symptoms. Older age, female sex, presence of comorbidity, alcohol use, low baseline hemoglobin level, fatigue at presentation, and prolonged hospital stay were found to increase the odds of developing PACS.

These risk factors provide valuable insights for healthcare professionals in identifying individuals who may be more susceptible to post-acute COVID-19 syndrome and require targeted monitoring and care. The research findings emphasize the critical importance of long-term healthcare management for COVID-19 survivors. Multidisciplinary approaches involving healthcare providers from various specialties will be crucial in providing holistic care to post-acute COVID-19 syndrome patients.

## Acknowledgments

We would like to thank our research participants who made this research complete. We are also grateful to the management and staff of St. Paul's Hospital Millennium Medical College for their cooperation.

## Author Contributions

**Conceptualization:** Tamrat Petros Elias, Tigist Tesfaye Gebremeskel, Abel Mureja Argaw, Biruk Mulugeta Belay.

**Data curation:** Tamrat Petros Elias, Tsegaye Wesenseged Gebreamlak, Abraham Sisay Abie, Bitaniya Petros Elias, Addis Aschenek Tenaw.

**Formal analysis:** Tamrat Petros Elias, Tigist Tesfaye Gebremeskel, Abraham Sisay Abie, Bitaniya Petros Elias, Addis Aschenek Tenaw, Biruk Mulugeta Belay.

**Investigation:** Tsegaye Wesenseged Gebreamlak, Tigist Tesfaye Gebremeskel, Binyam Lukas Adde, Bitaniya Petros Elias, Biruk Mulugeta Belay.

**Methodology:** Tamrat Petros Elias, Tigist Tesfaye Gebremeskel, Binyam Lukas Adde, Abraham Sisay Abie, Bitaniya Petros Elias, Abel Mureja Argaw, Addis Aschenek Tenaw, Biruk Mulugeta Belay.

**Project administration:** Tamrat Petros Elias, Tsegaye Wesenseged Gebreamlak, Tigist Tesfaye Gebremeskel, Binyam Lukas Adde, Abel Mureja Argaw.

**Software:** Tamrat Petros Elias, Tigist Tesfaye Gebremeskel, Binyam Lukas Adde, Bitaniya Petros Elias, Abel Mureja Argaw, Addis Aschenek Tenaw.

**Supervision:** Tamrat Petros Elias, Tsegaye Wesenseged Gebreamlak, Tigist Tesfaye Gebremeskel, Bitaniya Petros Elias, Abel Mureja Argaw.

**Validation:** Tamrat Petros Elias, Tsegaye Wesenseged Gebreamlak, Binyam Lukas Adde, Bitaniya Petros Elias, Abel Mureja Argaw.

**Writing – original draft:** Tamrat Petros Elias, Tigist Tesfaye Gebremeskel, Bitaniya Petros Elias.

**Writing – review & editing:** Tamrat Petros Elias, Tsegaye Wesenseged Gebreamlak, Binyam Lukas Adde, Abraham Sisay Abie, Abel Mureja Argaw, Addis Aschenek Tenaw, Biruk Mulugeta Belay.

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
