## [Decision Letter · Decision Letter 0]

24 Nov 2023

PONE-D-23-27462Assessment of the prevalence of post-acute COVID-19 syndrome and its associated risk factors among severe COVID-19 patients admitted to Millennium COVID-19 care center, Addis Ababa, EthiopiaPLOS ONE

Dear Dr. Elias,

Thank you for submitting your manuscript to PLOS ONE. After careful consideration, we feel that it has merit but does not fully meet PLOS ONE’s publication criteria as it currently stands. Therefore, we invite you to submit a revised version of the manuscript that addresses the points raised during the review process.

We look forward to receiving your revised manuscript.

Kind regards,

Reaz Mahmud, MBBS, FCPS (Medicine), MD (Neurology)

Academic Editor

PLOS ONE

Journal Requirements:

A clean copy of the edited manuscript (uploaded as the new *manuscript* file).

Additional Editor Comments:

Thank you for submitting your article to Plos One.

Using the term 'frequency' instead of 'prevalence' for non-population-based studies is advised. Please brief the background section and add a conclusion to the abstract. The headings for the rationale and objective should be omitted, as they are part of the introduction. Please briefly discuss PACS's pathophysiology in the introduction and include the operational definitions of PACS and severe COVID-19 in the methods section. The telephone interview guide should be included in the supplement, and the categorization of fatigue and cough in the telephonic interview should be mentioned. Please describe how you identified the patients from the hospital records, briefly discuss the sampling technique used, and explain how you created the sampling frame and interval. A patient selection flow chart should be provided in the results section. For quantitative data, please use median (IQR) instead of mean and perform non-parametric tests. In the first paragraph of the discussion section, summarize the findings rather than presenting data. Please discuss the generalizability and limitations of the study and their implications in the discussion section. Lastly, please rewrite some references according to Plos guidelines and copyedit the manuscript for language correction.

Reviewers' comments:

Reviewer's Responses to Questions

**Comments to the Author**

1. Is the manuscript technically sound, and do the data support the conclusions?

Reviewer #1: Yes

Reviewer #2: Yes

2. Has the statistical analysis been performed appropriately and rigorously? 

Reviewer #1: Yes

Reviewer #2: Yes

3. Have the authors made all data underlying the findings in their manuscript fully available?

Reviewer #1: No

Reviewer #2: Yes

4. Is the manuscript presented in an intelligible fashion and written in standard English?

Reviewer #1: Yes

Reviewer #2: Yes

5. Review Comments to the Author

Reviewer #1: The manuscript offers a detailed insight into the prevalence and risk factors associated with PACS in severe COVID-19 patients. The discussion effectively contextualizes the findings with previous literature, which is commendable. Key points include:

• Detailed exploration of the symptomatology, risk factors, and prevalence of PACS.

• Highlighting gender-based disparity in PACS presentation.

• Identifying significant associations like older age, prolonged hospital stay, and lifestyle factors.

Recommendations:

• Further elaborate on potential interventions or treatments for fatigue in PACS.

• Consider discussing the clinical implications of these findings for healthcare providers.

Minor grammatical errors were noticed in the manuscript. It would be beneficial for the authors to proofread the article for clarity.

Reviewer #2: The paper is well-written, and the methodology is robust. However, including additional details and clarifications in certain sections could enhance the overall quality and clarity of the research.

Here are some suggestions:

Introduction

Line 56, Consider adding a sentence or two explaining the potential impact of PACS on individuals, families, and society.

Line 59, 60. and 227, There is a discrepancy in the reference to prevalence percentages. It states "46% in Bangladesh to 81% in Italy," but later it mentions 87.4% in Italy. Please clarify this inconsistency.

Rationale

Line 74, Clarify the specific measures or interventions considered "effective management measures.

Methodology

Line 100, Provide more details about the adaptation process of the WHO Global COVID-19 Clinical Platform Case Report Form for Post-COVID conditions.

Results:

Line 128, Provide a percentage or proportion for the patients who died after hospital discharge (5.8%).

Lines 159-161, Consider providing more details on the health facility visits, such as the types of health facilities visited and the reasons for the visits.

Discussion:

Lines 200-210, The discussion is comprehensive and well-structured. Consider discussing potential limitations of the study, such as recall bias during telephone interviews.

Conclusion:

Lines 276-280, The conclusion summarizes the key findings well. Consider adding a brief statement about the implications of your findings for future research or public health interventions.

General comment:

Please review the manuscript for any potential grammatical errors to ensure the clarity and accuracy of the text before final submission.

6. PLOS authors have the option to publish the peer review history of their article (what does this mean?). If published, this will include your full peer review and any attached files.

Reviewer #1: **Yes: **Dr Victor Abiola Adepoju

Reviewer #2: No

---

## [Author Response · Author response to Decision Letter 0]

16 Dec 2023

PONE-D-23-27462

Assessment of the prevalence of post-acute COVID-19 syndrome and its associated risk factors among severe COVID-19 patients admitted to Millennium COVID-19 care center, Addis Ababa, Ethiopia

Dear Editor and Reviewers;

I hope this letter finds you well. I would like to express my gratitude for taking the time to review my manuscript. Your constructive feedback and comments have been invaluable in improving the quality and clarity of my work.

I would like to address the concerns and comments raised in your review. I have carefully considered each one of them and have made the necessary changes to the manuscript accordingly.

So please find the following comments and responses.

Journal Requirements:

- I have revised the manuscript according to PLOS ONE’s style requirements.

 - Raw data is deposited on Dryad: *https://datadryad.org/stash/share/X4rfM77vUc2yRwvIi0mI9FiyXbIla0CzHgJJdwJTv5g *https://doi.org/10.5061/dryad.fbg79cp2k

3. We suggest you thoroughly copyedit your manuscript for language usage, spelling, and grammar. 

 - The manuscript is now edited for language usage, spelling, and grammar. 

The name of the colleague or the details of the professional service that edited your manuscript.

- The manuscript is edited by Dr. Henok Fisseha. 

- The copy of the original manuscript with track changes is uploaded as supporting information. 

A clean copy of the edited manuscript (uploaded as the new *manuscript* file).

- A clean copy of the edited manuscript is uploaded. 

- A major revision is made to the reference section. One retracted reference is removed. Few new references are added. Finally, the reference list is reorganized based on order and formatted according to PLOS guidelines. 

Additional Editor Comments: 

Thank you for submitting your article to Plos One.

Using the term 'frequency' instead of 'prevalence' for non-population-based studies is advised. 

- Thank you for the comment. We used prevalence to measure the percentage of patients with PACS among the COVID-19 affected population. 

Please brief the background section and add a conclusion to the abstract. 

- Background section is briefed and conclusion added to the abstract. 

The headings for the rationale and objective should be omitted, as they are part of the introduction. 

- Headings for rationale and objective are omitted. 

Please briefly discuss PACS's pathophysiology in the introduction and include the operational definitions of PACS and severe COVID-19 in the methods section. 

- PACS pathophysiology is added to the introduction section and operational definitions of PACS and severe COVID-19 are added in the methods section. 

The telephone interview guide should be included in the supplement, and the categorization of fatigue and cough in the telephonic interview should be mentioned. 

- Both the English and Amharic versions of the telephonic interview guide are now attached in the supporting information now. We only used the study participants’ subjective response of whether the fatigue or cough that started during the acute illness persisted until the time of interview or not. We didn’t further categorized fatigue and cough. 

Please describe how you identified the patients from the hospital records, briefly discuss the sampling technique used, and explain how you created the sampling frame and interval.

- The procedures used to identify the patients from hospital records, the sampling technique used, how sampling frame and interval was created are now included in the methodology section. 

A patient selection flow chart should be provided in the results section. 

- Patient selection flow chat is included in the results section now. 

For quantitative data, please use median (IQR) instead of mean and perform non-parametric tests. 

- For quantitative data, except age, median (IQR) is used and Mann-Whitney U test is performed. 

In the first paragraph of the discussion section, summarize the findings rather than presenting data. 

- First paragraph of the discussion is now revised. Detailed data is removed a summary of the findings is included. 

Please discuss the generalizability and limitations of the study and their implications in the discussion section. 

- Limitations and generalizability of the study is included now in the discussion section. 

Lastly, please rewrite some references according to PLOS guidelines and copyedit the manuscript for language correction.

- Manuscript is edited for language correction and reference list is updated according to PLOS guidelines.

Reviewers' comments: 

Reviewer's Responses to Questions

Comments to the Author

1. Is the manuscript technically sound, and do the data support the conclusions?

Reviewer #1: Yes

Reviewer #2: Yes

- Thank you for the positive response. 

2. Has the statistical analysis been performed appropriately and rigorously?

Reviewer #1: Yes

Reviewer #2: Yes

- Thank you for the positive response. 

3. Have the authors made all data underlying the findings in their manuscript fully available?

Reviewer #1: No

Reviewer #2: Yes

- Raw data is deposited on the Dryad repository now. 

4. Is the manuscript presented in an intelligible fashion and written in Standard English?

Reviewer #1: Yes

Reviewer #2: Yes

- Thank you for the positive response. 

5. Review Comments to the Author

Reviewer #1: The manuscript offers a detailed insight into the prevalence and risk factors associated with PACS in severe COVID-19 patients. The discussion effectively contextualizes the findings with previous literature, which is commendable. Key points include:

• Detailed exploration of the symptomatology, risk factors, and prevalence of PACS.

• Highlighting gender-based disparity in PACS presentation.

• Identifying significant associations like older age, prolonged hospital stay, and lifestyle factors.

- Thank you for the positive response. 

Recommendations: 

• Further elaborate on potential interventions or treatments for fatigue in PACS.

- Effective interventions and treatments for fatigue in PACS is now included.

• Consider discussing the clinical implications of these findings for healthcare providers.

- The clinical implications of these findings for healthcare providers in now included in the conclusion section. 

Minor grammatical errors were noticed in the manuscript. It would be beneficial for the authors to proofread the article for clarity.

- The article is now edited for clarity and grammatical errors. 

Reviewer #2: The paper is well-written, and the methodology is robust.

- Thank you for the positive response. 

However, including additional details and clarifications in certain sections could enhance the overall quality and clarity of the research.

Here are some suggestions: 

Introduction

Line 56, Consider adding a sentence or two explaining the potential impact of PACS on individuals, families, and society.

- The Economic impact of PACS is now included with a new reference. 

Line 59, 60, and 227, there is a discrepancy in the reference to prevalence percentages. It states "46% in Bangladesh to 81% in Italy," but later it mentions 87.4% in Italy. Please clarify this inconsistency.

- Thank you for noticing the discrepancy. These are two separate studies conducted in Italy. Because the later one showed a higher prevalence, the first article is omitted. 

Rationale

Line 74, Clarify the specific measures or interventions considered "effective management measures.

- Specific measures or interventions are now briefly explained.

Methodology

Line 100, Provide more details about the adaptation process of the WHO Global COVID-19 Clinical Platform Case Report Form for Post-COVID conditions.

- The adaptation process is now discussed in the data collection tools and procedures sub section of the methodology. 

Results: 

Line 128, Provide a percentage or proportion for the patients who died after hospital discharge (5.8%).

- Percentage of patients who died after hospital discharge is now include in the results section.

Lines 159-161, consider providing more details on the health facility visits, such as the types of health facilities visited and the reasons for the visits.

- The health facilities are either clinic, health center, or hospital. The most common reason (59%) for the health facility visit was for follow-up of chronic disease like diabetes or hypertension, and 15.7% of the reasons for hospital visit after discharge were not feeling well.

Discussion: 

Lines 200-210, the discussion is comprehensive and well-structured. Consider discussing potential limitations of the study, such as recall bias during telephone interviews.

- Limitations of the study are now included at the end of the results section. 

Conclusion: 

Lines 276-280, The conclusion summarizes the key findings well. Consider adding a brief statement about the implications of your findings for future research or public health interventions.

- The implications of the findings for future research and public health intervention are now included at the end of the discussion section and the conclusion part. 

General comment: 

Please review the manuscript for any potential grammatical errors to ensure the clarity and accuracy of the text before final submission.

- The manuscript is now checked for grammatical errors. 

---

## [Decision Letter · Decision Letter 1]

3 Jan 2024

PONE-D-23-27462R1Assessment of the prevalence of post-acute COVID-19 syndrome and its associated risk factors among severe COVID-19 patients admitted to Millennium COVID-19 care center, Addis Ababa, EthiopiaPLOS ONE

Dear Dr. Elias,

Thank you for submitting your manuscript to PLOS ONE. After careful consideration, we feel that it has merit but does not fully meet PLOS ONE’s publication criteria as it currently stands. Therefore, we invite you to submit a revised version of the manuscript that addresses the points raised during the review process.

We look forward to receiving your revised manuscript.

Kind regards,

Academic Editor

PLOS ONE

Journal Requirements:

**Additional Editor Comments:**

Please revise and shorten the Title.

There is no need to specify the city name in the Title unless necessary.

Reviewers' comments:

Reviewer's Responses to Questions

**Comments to the Author**

1. If the authors have adequately addressed your comments raised in a previous round of review and you feel that this manuscript is now acceptable for publication, you may indicate that here to bypass the “Comments to the Author” section, enter your conflict of interest statement in the “Confidential to Editor” section, and submit your "Accept" recommendation.

Reviewer #1: All comments have been addressed

Reviewer #2: All comments have been addressed

2. Is the manuscript technically sound, and do the data support the conclusions?

Reviewer #1: Yes

Reviewer #2: Yes

3. Has the statistical analysis been performed appropriately and rigorously? 

Reviewer #1: Yes

Reviewer #2: Yes

4. Have the authors made all data underlying the findings in their manuscript fully available?

Reviewer #1: Yes

Reviewer #2: Yes

5. Is the manuscript presented in an intelligible fashion and written in standard English?

Reviewer #1: Yes

Reviewer #2: Yes

6. Review Comments to the Author

Reviewer #1: The authors have addressed all the issues raised by the reviewers and current iteration merit publication after some editorial polishing of the paper. Good job

Reviewer #2: (No Response)

7. PLOS authors have the option to publish the peer review history of their article (what does this mean?). If published, this will include your full peer review and any attached files.

Reviewer #1: **Yes: **No

Reviewer #2: No

---

## [Author Response · Author response to Decision Letter 1]

4 Jan 2024

PONE-D-23-27462R1

Assessment of the prevalence of post-acute COVID-19 syndrome and its associated risk factors among severe COVID-19 patients admitted to Millennium COVID-19 Care Center, Addis Ababa, Ethiopia

Dear Editor and Reviewers;

I hope this letter finds you well. I would like to express my gratitude for taking the time to review my manuscript. Your constructive feedback and comments have been invaluable in improving the quality and clarity of my work.

I would like to address the concerns and comments raised in your review. I have carefully considered each one of them and have made the necessary changes to the manuscript accordingly.

So please find the following comments and responses.

Journal Requirements:

1. Please review your reference list to ensure that it is complete and correct.

- I've re-checked the reference list and found it to be complete and correct. 

2. If you have cited papers that have been retracted, please include the rationale for doing so in the manuscript text, or remove these references and replace them with relevant current references.

- No retracted papers are cited in the current submission. 

3. Any changes to the reference list should be mentioned in the rebuttal letter that accompanies your revised manuscript. If you need to cite a retracted article, indicate the article’s retracted status in the References list and also include a citation and full reference for the retraction notice.

- No changes are made to the reference list. 

Additional Editor Comments: 

Please revise and shorten the Title.

There is no need to specify the city name in the Title unless necessary.

Thank you for the comment. The title is now revised to “Determinants of Post-acute COVID-19 Syndrome among hospitalized severe COVID-19 Patients: A 2-year follow-up study”

Reviewers' comments: 

Reviewer's Responses to Questions

Comments to the Author

1. If the authors have adequately addressed your comments raised in a previous round of review and you feel that this manuscript is now acceptable for publication, you may indicate that here to bypass the “Comments to the Author” section, enter your conflict of interest statement in the “Confidential to Editor” section, and submit your "Accept" recommendation.

Reviewer #1: All comments have been addressed

Reviewer #2: All comments have been addressed

- Thank you for the positive response. 

2. Is the manuscript technically sound, and do the data support the conclusions?

Reviewer #1: Yes

Reviewer #2: Yes

- Thank you for the positive response. 

3. Has the statistical analysis been performed appropriately and rigorously?

Reviewer #1: Yes

Reviewer #2: Yes

- Thank you for the positive response. 

4. Have the authors made all data underlying the findings in their manuscript fully available?

Reviewer #1: Yes

Reviewer #2: Yes

- Thank you for the positive response. 

5. Is the manuscript presented in an intelligible fashion and written in standard English?

Reviewer #1: Yes

Reviewer #2: Yes

- Thank you for the positive response. 

6. Review Comments to the Author

Reviewer #1: The authors have addressed all the issues raised by the reviewers and current iteration merit publication after some editorial polishing of the paper. Good job

Reviewer #2: (No Response)

- Thank you for the positive response.

---

## [Decision Letter · Decision Letter 2]

25 Jan 2024

Determinants of Post-acute COVID-19 Syndrome among hospitalized severe COVID-19 Patients: A 2-year follow-up study

PONE-D-23-27462R2

Dear Dr. Elias,

We’re pleased to inform you that your manuscript has been judged scientifically suitable for publication and will be formally accepted for publication once it meets all outstanding technical requirements.

Kind regards,

Academic Editor

PLOS ONE

Additional Editor Comments (optional):

Reviewers' comments:

Reviewer's Responses to Questions

**Comments to the Author**

1. If the authors have adequately addressed your comments raised in a previous round of review and you feel that this manuscript is now acceptable for publication, you may indicate that here to bypass the “Comments to the Author” section, enter your conflict of interest statement in the “Confidential to Editor” section, and submit your "Accept" recommendation.

Reviewer #2: All comments have been addressed

2. Is the manuscript technically sound, and do the data support the conclusions?

Reviewer #2: Yes

3. Has the statistical analysis been performed appropriately and rigorously? 

Reviewer #2: Yes

4. Have the authors made all data underlying the findings in their manuscript fully available?

Reviewer #2: Yes

5. Is the manuscript presented in an intelligible fashion and written in standard English?

Reviewer #2: Yes

6. Review Comments to the Author

Reviewer #2: The authors have diligently addressed all concerns. They confirm the reference list's accuracy and no citation of retracted papers. Appreciate their thoroughness.

7. PLOS authors have the option to publish the peer review history of their article (what does this mean?). If published, this will include your full peer review and any attached files.

Reviewer #2: No

---

## [Editor Report · Acceptance letter]

29 Apr 2024

PONE-D-23-27462R2 

PLOS ONE

Dear Dr. Elias, 

I'm pleased to inform you that your manuscript has been deemed suitable for publication in PLOS ONE. Congratulations! Your manuscript is now being handed over to our production team.

Kind regards, 

on behalf of

Dr. Robert Jeenchen Chen 

Academic Editor

PLOS ONE